# On the Diversity of Semiochemicals of the Pygidial Gland Secretions of Subterranean Ground Beetles (Coleoptera: Carabidae)

**Sofija Vranić [1], Ljubodrag Vujisić [2,*], Nikola Vesović [1], Marina Todosijević [2], Miloš Pavićević [3], Dejan Radović [4] and Srećko Ćurčić [1]**

1   Faculty of Biology, Institute of Zoology, University of Belgrade, Studentski Trg 16, 11000 Belgrade, Serbia
2   Faculty of Chemistry, University of Belgrade, Studentski Trg 12–16, 11000 Belgrade, Serbia
3   Biospeleological Society of Montenegro, Cara Lazara 22, 81000 Podgorica, Montenegro
4   Faculty of Security Studies, University of Belgrade, Gospodara Vučića 50, 11118 Belgrade, Serbia
*   Correspondence: ljubaw@chem.bg.ac.rs; Tel.: +381-11-3336-899

**Abstract:** Pygidial glands are of great importance to ground beetles for defense against predators, especially for the species that live in subterranean habitats. The purpose of our study is to better understand the chemistry of the pygidial gland secretions of subterranean ground beetles, as well as the function and structure of the glands. We studied both the chemical composition of the pygidial gland secretion and morphology of the glands in adults of the troglophilic ground beetle species *Laemostenus* (*Antisphodrus*) *cavicola* (Schaum, 1858). The chemical composition of its defensive secretion was revealed using gas chromatography-mass spectrometry (GC-MS), while pygidial gland morphology of the beetle was investigated using bright-field microcopy. In total, seven chemical compounds were detected in the secretion mixture. Formic acid was the most dominant compound, followed by dodecyl acetate and undecane. Other chemicals were present in minor amounts. The morphological structure of the pygidial glands of *L.* (*A.*) *cavicola* was compared with the structure of the glands of the related congeneric troglophilic species *Laemostenus* (*Pristonychus*) *punctatus* (Dejean, 1828). Summary data on the semiochemicals that have been recorded so far in subterranean ground beetle species are presented, and the differences in the chemical composition of the secretions between and among troglobitic and troglophilic species are discussed. So far, forty-four compounds have been detected in four subterranean ground beetle species (two troglobites belonging to the tribe Trechini and two troglophiles belonging to the tribe Sphodrini). The results of this study indicate the great diversity of chemicals in the pygidial gland secretions of subterranean ground beetles.

**Keywords:** carabid beetles; Platyninae; Trechinae; defensive glands; secretion mixtures; gas chromatography-mass spectrometry





## 1. Introduction

Chemically-mediated communication in insects is well documented [1,2]. Small organic compounds, also known as semiochemicals, act as chemical cues. These are often divided into three groups: pheromones, allomones and kairomones. While the chemicals of the former group are recognized by conspecifics, the chemical compounds of the remaining two groups mediate interspecific interactions [2,3].

Ground beetles (Carabidae) release a wide variety of chemicals from a pair of abdominal glands, called pygidial glands. Each pygidial gland comprises secretory lobes (in which the production of defensive secretions takes place), a reservoir (for the storage of defensive secretions) and transportation elements. The latter include radial collecting canals that carry glandular secretions from individual secretory lobes and merge into a main collecting canal that leads secretion from the secretory lobes to the reservoir, as well as an efferent duct via which secretion is released into the environment [4,5].

The pygidial gland secretions of ground beetles are primarily regarded as allomones as they exhibit deterrent, toxic and irritant properties serving in the defense against predators, yet additional functions have also been taken into consideration [1,2,6]. Some chemical products of pygidial glands have been proven to have antimicrobial properties [7–9]. Furthermore, it was hypothesized that certain chemical compounds play a role in sexual communication or serve as alarm pheromones in conspecifics [2,10]. To date, pygidial gland secretions have been researched in about 500 species of ground beetles from all over the world [11,12].

Species of the tribe Sphodrini were rarely chemoecologically studied in the past. Schildknecht et al. [13] first analyzed the defensive secretions of two European species of the genus *Calathus* Bonelli, 1810: *Calathus* (*Calathus*) *fuscipes* (Goeze, 1777) and *C.* (*Neocalathus*) *melanocephalus* (Linnaeus, 1758). Following that, Will et al. [14] carried out a study on the defensive secretion of the Nearctic *Calathus* (*Neocalathus*) *ruficollis* Dejean, 1828. Finally, Vesović et al. [15] analyzed the chemical composition of the defensive secretion of the troglophilic *Laemostenus* (*Pristonychus*) *punctatus* (Dejean, 1828). In the same paper [15], the secretions of two troglobitic ground beetle taxa of the tribe Trechini, *Duvalius* (*Paraduvalius*) *milutini* S. Ćurčić, Vrbica, Antić & B. Ćurčić, 2014 and *Pheggomisetes globiceps ninae* S. Ćurčić, Schönmann, Brajković, B. Ćurčić & Tomić, 2004 (Serbian stenoendemics), were chemoecologically investigated. That was the first and, so far, the only study to provide data on the semiochemicals of the pygidial gland secretion mixtures from both cave-dwelling ground beetles (in total, three taxa: one troglophile and two troglobites) and representatives of the tribe Trechini [15].

Caves and pits are subterranean habitats characterized by stable ecological conditions, which significantly differ from those in the surrounding surface habitats [16]. Such conditions led to the development of unique adaptations in the organisms inhabiting those habitats. Some of the adaptations are directed towards the reduction of morphological structures that are not of use in subterranean habitats. For instance, eye and hind wing reduction is documented in many cave-dwelling insects [16]. Considering the aforementioned trend of reduction, as well as decreasing predation pressure, Vesović et al. [15] suggested that the defensive secretions of subterranean ground beetles would be simplified in adapted (troglobitic) species compared to less adapted (troglophilic) ones [15]. However, the results of this study were contrary to that hypothesis. No further studies on subterranean ground beetles' defensive secretions have been conducted in the meantime.

The pygidial glands of representatives of the tribe Sphodrini have not been the focus of many investigations in the past. Gland morphology of *Calathus* (*Neocalathus*) *ambiguus* (Paykull, 1790) and *Laemostenus* (*Pristonychus*) *terricola* (Herbst, 1784) was briefly mentioned by Forsyth [4], with no detailed description or measurements provided. The first in-depth morphological study on the pygidial glands of Sphodrini was carried out by Nenadić et al. [8] for the species *L.* (*P.*) *punctatus*. Later, scanning electron microscopy and nonlinear microscopy were introduced as methods that enabled investigations of the ultrastructure of the pygidial glands of the mentioned ground beetle species [17].

In the present study, we chose to chemoecologically investigate the defensive secretion mixture and study pygidial gland morphology of the ground beetle species *Laemostenus* (*Antisphodrus*) *cavicola* (Schaum, 1858), which belongs to the tribe Sphodrini (Figure 1). The mentioned species is a troglophile, it is adapted to complete its life cycle in caves, but can also be found outside of caves. Contrary to troglophiles, troglobites are strictly bound to subterranean habitats, such as caves and pits. The abovementioned troglophilic species is distributed in southern and southeastern Europe, mainly on the Balkan Peninsula [18]. This species was not previously investigated in terms of chemical ecology or pygidial gland morphology.

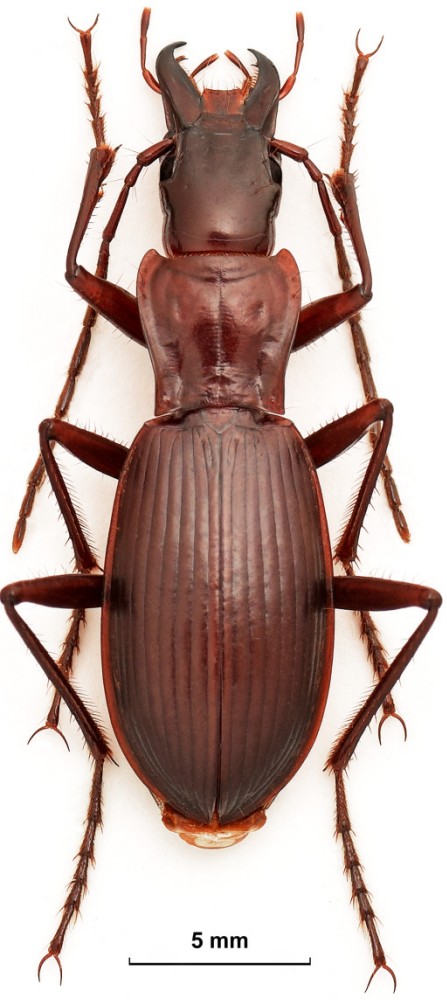

**Figure 1.** Habitus (dorsal view) of the adult specimen of *L.* (*A.*) *cavicola*. Photo N. Vesović.

Our aims were to: (i) identify chemical compounds in the defensive pygidial gland secretion of *L.* (*A.*) *cavicola*; (ii) examine pygidial gland morphology of the same species; (iii) compare the chemical composition of the secretion and the structure of the glands with related subterranean species; (iii) summarize data on all semiochemicals present so far in subterranean ground beetles; and (iv) discuss the differences in the chemical composition of the secretions between troglophilic and troglobitic representatives of ground beetles.

## 2. Materials and Methods

### 2.1. Sample Collection

Eight adult specimens (four males and four females) of *L.* (*A.*) *cavicola* were collected in the Grbočica Cave, village of Trnovo, area of Crmnica, close to the settlement of Virpazar, southern Montenegro. Ground beetle individuals were manually collected by S. Ćurčić and M. Pavićević on 30 September 2018. These were then placed in a portable chamber along with moist substrate from the collecting site. Temperature was kept at a constant level (10 °C). The sufficient level of humidity was maintained by occasional spraying of water. The ground beetles were fed on earthworms.

### 2.2. Chemical Analyses of Pygidial Gland Secretion

Sample preparation for gas chromatography-mass spectrometry (GC-MS) was conducted at room temperature in the laboratory of the University of Belgrade - Faculty of Chemistry (Belgrade, Serbia). Each individual beetle of the same sex was stimulated to discharge its defensive pygidial gland secretion by squeezing the tip of its abdomen and

by pinching the legs with a tweezers into a single 12-mL glass vial with dichloromethane (0.5 mL) as a solvent (Merck, Darmstadt, Germany). The samples were subjected to GC-MS analyses immediately after their preparation. Samples were analyzed on the GC-MS system (Agilent 7890A–5975C, Agilent Technologies, Santa Clara, CA, USA) in splitless mode (with 1 μL injection volume) on a polar HP-INNOWax capillary column (30 m × 0.32 mm × 0.25 μm). Oven temperature was linearly programed in the range of 40–240 °C at a rate of 10 °C min$^{-1}$, with a final 10-min hold. The electron ionization (EI) (70 eV) mass spectral range was 40–550 *m/z*. Compounds were identified by comparison with commercially available NIST 17 and Willey 07 mass spectral libraries containing more than half a million spectra. In addition, all compounds were characterized by retention indices (RIs) obtained from the corresponding series of *n*-alkanes analyzed under the same chromatographic conditions immediately after the sample run. RIs obtained on standard polar capillary columns were compared with the available literature data from NIST Chemistry WebBook and PubChem (Table 1). The relative mass percentages of the identified chemicals were calculated from the corresponding areas of the GC-MS peaks. This is particularly important if analyzed compounds have different polarities and/or concentrations that cause peak broadening, because in that case measuring peak heights instead of areas would be misleading (Figure 2, Table 1).

**Table 1.** The chemical composition of the pygidial gland secretion of *L.* (*A.*) *cavicola* analyzed by GC-MS on a polar column (HP-INNOWax 30 m × 0.32 mm × 0.25 μm).

| Peak | Rt (min) | Compound | RI | RI$_{lit}$ | Relative Percentage (%) |
|------|----------|----------|-----|-----------|------------------------|
| 1 | 4.16 | Undecane | 1100 | 1100 | 27.0 |
| 2 | 8.79 | Acetic acid | 1465 | 1400–1498 | 0.5 |
| 3 | 9.38 | Formic acid | 1508 | 1470–1544 | 37.4 |
| 4 | 13.81 | Dodecyl formate | 1858 | - | 0.8 |
| 5 | 14.31 | Dodecyl acetate | 1901 | 1876–1900 | 33.5 |
| 6 | 16.54 | 1-Tetradecyl acetate | 2104 | 2062–2106 | 0.1 |
| 7 | 26.16 | Palmitic acid | 2935 | 2871–2954 | 0.7 |

RI—retention indices calculated from the GC-MS retention times of a series of *n*-alkanes obtained under the same chromatographic conditions; RI$_{lit}$—retention indices from summarized NIST Chemistry WebBook and PubChem literature data obtained on standard polar capillary columns; Rt—retention time.

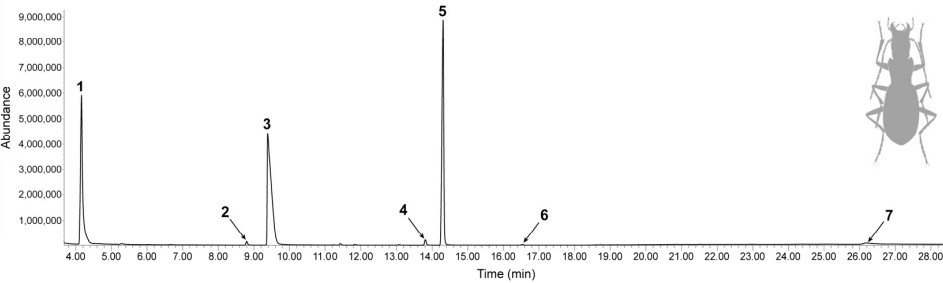

**Figure 2.** Gas chromatography-mass spectrometry (GC-MS) total ion chromatogram of the dichloromethane pygidial gland secretion extracts from the adults of *L.* (*A.*) *cavicola*. Ordinal numbers above peaks correspond to numbers in Table 1.

### 2.3. Morphological Analyses

A pair of pygidial glands were carefully extracted in 70% ethanol. All morphological features of the glands were observed and precisely measured. Gland structures were photographed with a Nikon SMZ800N stereomicroscope equipped with a Nikon DS-Fi2 digital camera (Nikon Corp., Tokyo, Japan). The measuring of different parts of the pygidial glands was conducted using a Nikon DS-L3 control unit (Nikon Corp., Tokyo, Japan).

The analyzed ground beetle specimens were deposited in the entomological collection of the Institute of Zoology, University of Belgrade-Faculty of Biology (Belgrade, Serbia).

## 3. Results

*3.1. Chemical Composition of Pygidial Gland Secretion*

In total, seven chemical compounds were detected in the pygidial gland secretion mixture of *L.* (*A.*) *cavicola* (Table 1). The secretion extract of the mentioned species contained one hydrocarbon (alkane), three carboxylic acids (two short-chain and one long-chain) and three esters. The most abundant compound in the mixture was formic acid (37.4%), followed by dodecyl acetate (33.5%) and undecane (27.0%). The remaining compounds (dodecyl formate, palmitic acid, acetic acid and 1-tetradecyl acetate) were found in minor amounts (each with less than 1%) (Figure 2, Table 1). No qualitative and quantitative differences in the chemical composition of the secretion were observed between the sexes of *L.* (*A.*) *cavicola*.

In comparison to the only previously analyzed congener, *L.* (*P.*) *punctatus* (at the same time, this is the only chemoecologically studied troglophilic species of ground beetles to date), *L.* (*A.*) *cavicola* had a simpler secretion mixture, with only seven compounds detected [vs. thirteen compounds found in *L.* (*P.*) *punctatus*] [15]. However, certain chemoecological features of the defensive secretions appeared to be similar among these related species. The presence of formic acid, acetic acid, alkanes, esters and fatty acids was recorded in both species. Formic acid, dodecyl acetate and undecane were major compounds in both species, even though *L.* (*P.*) *punctatus* had a somewhat lower percentage of the former compound [19.4% vs. 37.4% in *L.* (*A.*) *cavicola*] [15]. Acetic acid and fatty acids [caproic, palmitic, stearic and oleic acids in *L.* (*P.*) *punctatus* vs. only palmitic acid in *L.* (*A.*) *cavicola*] were minor constituents of the secretion mixtures in both species. While the secretion of *L.* (*P.*) *punctatus* possessed only long-chain acetates (decyl acetate, undecyl acetate and dodecyl acetate) in its pygidial gland secretion, the one of *L.* (*A.*) *cavicola* was distinguished by the presence of dodecyl formate. The defensive secretion of *L.* (*P.*) *punctatus* was characterized by the greater diversity of chemicals in terms of the recorded number of alkanes [3 vs. 1 in *L.* (*A.*) *cavicola*] and fatty acids [4 vs. 1 in *L.* (*A.*) *cavicola*]. Finally, the secretion of *L.* (*A.*) *cavicola* lacked an alcohol (1-dodecanol) and caproic acid, while these chemicals were present in the secretion of *L.* (*P.*) *punctatus* [15].

Comparing the semiochemical content of the secretions of troglophilic Sphodrini with compounds in the secretion samples of surface-dwelling Sphodrini, many similarities could be noted. The mixture of formic acid, various alkanes and long-chain esters, as well as the presence of acetic acid and different fatty acids, is typical for the species of both groups [13–15]. Most troglophilic species of Sphodrini can often be found in surface habitats, so it is not surprising that the chemical composition of their secretions is similar to the content recorded in species that live in outdoor habitats. However, it seems that the production of alcohols and caproic acid is restricted only to cave-dwelling Sphodrini [15].

Two chemoecologically investigated troglobitic ground beetle species significantly differed in regards to the content of their defensive secretions. It was proven that *Pheggomisetes globiceps* Buresch, 1925 is capable of producing an aldehyde (benzaldehyde), a phenol (*p*-cresol), hydrocarbons (alkanes and alkenes) and carboxylic acids in its pygidial glands. In contrast to that, the secretion of *D.* (*P.*) *milutini* was characterized by the presence of carboxylic acids alone, including those of variable chain length (four medium-chain and four long-chain) and an aromatic one (benzoic acid). Nevertheless, the defensive secretions of these two species shared some features. Benzoic, caproic and all four long-chain fatty acids, present in *D.* (*P.*) *milutini*, were also found in *P. globiceps*. Three medium-chain fatty acids (pelargonic, capric and lauric) were only present in *D.* (*P.*) *milutini* [15].

Comparison between troglophilic and troglobitic ground beetles in terms of their defensive compounds meets certain difficulties, as the analyzed species of both groups belong to different subfamilies (troglophiles to Platyninae, and troglobites to Trechinae). However, it is worthwhile to compare certain chemoecological features of both groups' semiochemicals. While representatives of both groups possess hydrocarbons, long-chain fatty acids and various aliphatic low-molecular-weight carboxylic acids in their defensive secretions, some additional compounds (e.g., aldehydes, *p*-cresol and aromatic carboxylic

acids) can be found only in troglobitic taxa [15]. The overall diversity of hydrocarbons, low-molecular-weight carboxylic acids and fatty acids was greater in troglobites. In total, seventeen hydrocarbons, sixteen carboxylic acids and six fatty acids were isolated from troglobitic taxa. In the two analyzed troglophilic species, each of these three groups of compounds contained a total of three chemicals [15]. On the other hand, formic acid was not detected in the troglobites analyzed so far [15]. Troglobitic ground beetle taxa also lacked esters and alcohols. According to Vesović et al. [15], the observed differences between troglobites and troglophiles may be intergeneric, as ground beetles are characterized by a high level of conservation, so their defensive secretions should not be greatly affected by the existing selective pressures. Out of the analyzed troglobitic ground beetles, *P. globiceps* possessed the largest number of compounds (32). On the other hand, *D. (P.) milutini* contained only nine carboxylic acids in its pygidial gland secretion. Two troglophilic *Laemostenus* species were more alike regarding the number of detected compounds [*L. (A.) cavicola* and *L. (P.) punctatus* had seven and thirteen compounds, respectively]. Even though the pygidial gland secretion of *L. (A.) cavicola* had the smallest number of compounds, it possessed all classes of organic compounds characteristic for representatives of the tribe Sphodrini (short-chain carboxylic acids, alkanes, esters and fatty acids). Considering all analyzed cave-dwelling ground beetle species (two troglophiles and two troglobites), forty-four compounds were isolated from their pygidial gland secretion extracts. Hydrocarbons and carboxylic acids (nineteen and seventeen, respectively) were the most numerous compounds. Esters, which were limited to troglophilic species, were less abundant (five). Aldehydes, phenols and alcohols were represented by a single compound each [15]. A list of all chemical compounds detected in the pygidial gland secretions of subterranean ground beetle species studied to date is shown in Table 2.

### 3.2. Pygidial Gland Morphology

The paired pygidial glands of *L. (A.) cavicola* are composed of clustered secretory lobes, which are white and spherical (Figure 3A,C). The diameter of secretory lobes varies between 210 and 300 μm. The number of secretory units (lobes) per cluster ranges between 25 and 30. The main collecting canal is 2 cm long and 50–80 μm wide (Figure 3A). The diameter of its lumen is 20–30 μm. The main collecting canal enters the basal part of the reservoir near the beginning of the efferent duct (Figure 3B). The reservoir is elongated, ellipsoidal in shape and medially flattened at its inner margin. The lumen of the reservoir cannot be seen due to the thickness of its well-developed muscle wall (Figure 3B). The length of the reservoir is 1.97 mm and its width is 0.88 mm. It sharply and distally narrows, forming a 0.18-mm wide and 1.3-mm long efferent duct through which secretion is released to the outside (Figure 3B).

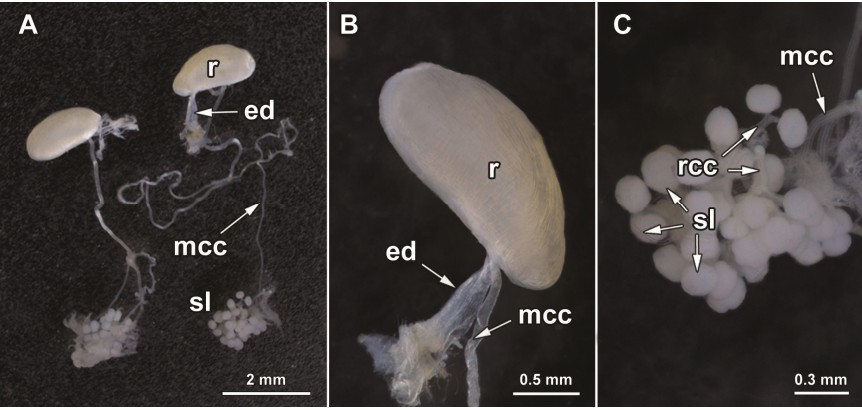

**Figure 3.** Pygidial gland morphology of *L. (A.) cavicola*: (**A**) a pair of the glands; (**B**) a reservoir with the distal part of a main collecting canal and an efferent duct; (**C**) a cluster of secretory lobes. Abbreviations: ed—efferent duct; mcc—main collecting canal; r—reservoir; rcc—radial collecting canal; sl—secretory lobe. Photo N. Vesović.

**Table 2.** The chemical composition of the pygidial gland secretions of subterranean ground beetle species (the first two are troglophilic, while the latter two are troglobitic) analyzed to date after [15].

| Compound | *L. (A.) cavicola* | *L. (P.) punctatus* | *D. (P.) milutini* | *P. globiceps* |
|---|---|---|---|---|
| Undecane | * | * | | * |
| Tridecane | | | | * |
| Acetic acid | * | * | | * |
| Formic acid | * | * | | |
| Benzaldehyde | | | | * |
| Propionic acid | | | | * |
| Isobutyric acid | | | | * |
| Butyric acid | | | | * |
| Isovaleric acid | | | | * |
| Decyl acetate | | * | | |
| Undecyl acetate | | * | | |
| Dodecyl formate | * | | | |
| Dodecyl acetate | * | * | | |
| 1-Tetradecyl acetate | * | | | |
| Isocaproic acid | | | | * |
| 1-Dodecanol | | * | | |
| Caproic acid | | * | * | * |
| *p*-Cresol | | | | * |
| Pelargonic acid | | | * | |
| Capric acid | | | * | |
| 7-Hexyldocosane | | * | | |
| 9-Methyltetracosane | | * | | |
| Benzoic acid | | | * | * |
| Lauric acid | | | * | |
| Pentacosane | | | | * |
| Pentacosene | | | | * |
| 3-Ethyltetracosane | | | | * |
| Hexacosane | | | | * |
| Myristic acid | | | * | * |
| 11-Methylheptacosane | | | | * |
| Heptacosene | | | | * |
| Heptacosadiene isomer 1 | | | | * |
| Heptacosadiene isomer 2 | | | | * |
| Octacosane | | | | * |
| Palmitic acid | * | * | * | * |
| Nonacosene | | | | * |
| Nonacosadiene isomer 1 | | | | * |
| Nonacosadiene isomer 2 | | | | * |
| Nonacosadiene isomer 3 | | | | * |
| Nonacosatetraene | | | | * |
| Nonacosapentaene | | | | * |
| Stearic acid | | * | * | * |
| Oleic acid | | * | * | * |
| Linoleic acid | | | | * |

*—The presence of compounds in the species.

Each pygidial gland of *L.* (*A.*) *cavicola* comprises 30–35 secretory lobes, which is significantly fewer than the number of lobes reported by other authors (60–70 lobes per cluster) for *L.* (*P.*) *terricola* and *L.* (*P.*) *punctatus* [4,8]. The number of lobes in *L.* (*A.*) *cavicola* is also greater than the number of the same structures (13) reported in *C.* (*N.*) *ambiguus*, which belongs to the same tribe (Sphodrini) [4]. Forsyth [4] reported the presence of only 12 secretory lobes in representatives of the tribe Platynini that belongs to the same subfamily (Platyninae). On the other hand, 13 Japanese species of Platyninae have a great number of secretory lobes (50 or more) [19]. According to the results of the aforementioned studies, it seems that the number of lobes varies within the entire subfamily Platyninae, but also among its taxa. The spherical shape of the lobes in *L.* (*A.*) *cavicola* indicates the presence of carboxylic acids as dominant compounds [19], which was proven by chemical analyses. The average size of a single lobe was somewhat greater in *L.* (*A.*) *cavicola* (210–300 μm) compared to *L.* (*P.*) *punctatus* (150–200 μm) (Table 3) [8]. The presence of a long main collecting canal is a common feature for all Platyninae [4,8]. In *L.* (*A.*) *cavicola*, it is 20 mm long, which is longer than the same structure in *L.* (*P.*) *punctatus* (10–15 mm) (Table 3) [8]. Interestingly, even though it is shorter, the main collecting canal in *L.* (*P.*) *punctatus* has a greater diameter (120 μm) than the one in *L.* (*A.*) *cavicola* (50–80 μm) [17]. The muscle-coated reservoir in *L.* (*A.*) *cavicola* is elongated, with the top and the base almost equal, unlike those that occur in the related species *L.* (*P.*) *punctatus*. Furthermore, the inner side of the reservoir in *L.* (*A.*) *cavicola* lacks a depression, which is very conspicuous in *L.* (*P.*) *punctatus* [8]. The lengths of the reservoirs in *L.* (*A.*) *cavicola* and *L.* (*P.*) *punctatus* are nearly identical (1.97 and 2.00 mm, respectively). However, the reservoir of *L.* (*P.*) *punctatus* is almost twice as wide as the one in *L.* (*A.*) *cavicola* (1.50 and 0.88 mm, respectively) (Table 3) [8]. The entering points of the main collecting canal and the efferent duct in *L.* (*A.*) *cavicola* are close to each other, as has been reported for other Platyninae [4,8]. The efferent duct in *L.* (*A.*) *cavicola* is about three times shorter than the one in *L.* (*P.*) *punctatus* (1.30 and 4.00 mm, respectively) (Table 3) [8]. A description of the main collecting canal and the efferent duct was not given in detail in previous studies [4,8]. Furthermore, these structures are conservative parts of pygidial glands and exhibit less variability within ground beetles, and are therefore less informative [4].

**Table 3.** Comparative measurements of different pygidial glands structures in *L.* (*A.*) *cavicola* and *L.* (*P.*) *punctatus* after [8]. All measurements are expressed in millimeters.

| Species | Gland Structure | | | | | |
|---|---|---|---|---|---|---|
| | Secretory Lobes Diameter | Main Collecting Canal | | Reservoir | | Efferent Duct |
| | | Length | Width | Length | Width | Length |
| *L.* (*A.*) *cavicola* | 0.21–0.30 | 20 | 0.05–0.08 | 1.97 | 0.88 | 1.30 |
| *L.* (*P.*) *punctatus* | 0.15–0.20 | 10–15 | 0.12 | 2.00 | 1.50 | 4.00 |

## 4. Discussion

The great diversity of hydrocarbons in the analyzed troglobitic ground beetles might not be related to the colonization of cave habitats. Since a similar situation was found in some epigean Bembidiini [*Bembidion* (*Peryphanes*) *deletum* Audinet-Serville, 1821 and *B.* (*Peryphus*) *subcostatum* (Motschulsky, 1850)], it might rather be considered as a characteristic of the subfamily Trechinae [20], where chemoecologically analyzed troglobites of the tribe Trechini also belong. The pygidial gland secretions of the mentioned species of Bembidiini contained a complex secretion mixture of low-molecular-weight carboxylic acids, as was the case with troglobitic representatives of the same subfamily. Interestingly, in other chemically tested species of the subfamily Trechinae, such a diversity of compounds has not been found. There is also the possibility that some of the detected compounds do not originate from the secretions of the pygidial glands, but from the cuticle, which depends on the sampling method that was used. At this point, any comprehensive conclusions cannot be made due to the small number of chemically tested species and individuals of the subfamily Trechinae. Furthermore, the absence of formic acid in troglobites could be attributed to the lack of need for aggressive substances under conditions of low predation risk, but

also to the fact that this chemical is rare among the members of the subfamily Trechinae (it has never been detected in species of the tribe Trechini to date) [20]. Other carboxylic acids are also considered aggressive irritants [1]. As mentioned before, both chemoecologically analyzed troglobites are characterized by the great diversity of chemicals belonging to that class of organic compounds. Various hydrocarbons (as in the case of *P. globiceps*) or esters (as in two troglophilic *Laemostenus* species), which are known to increase the repellent properties of the secretions, certainly represent effective predator deterrents [15]. The presence of benzaldehyde in the secretion of *P. globiceps* is fairly unusual as the distribution of the compound in the pygidial gland secretions of ground beetles was thought to be limited only to representatives of tiger beetles (subfamily Cicindelinae) [21–23]. At the same time, the question of its origin arises. Tiger beetles are assumed to utilize a cyanogenic pathway that is known to occur in other benzaldehyde-secreting species [21]. However, cyanogenesis is not widely distributed, and it would be informative to chemoecologically investigate related species in order to establish the distributional pattern of benzaldehyde in ground beetles.

Nenadić et al. [8,9] and Dimkić et al. [17] proved certain antimicrobial properties of the defensive secretion of the troglophilic species *L. (P.) punctatus*, which is a relative of *L. (A.) cavicola*. Carboxylic acids and an alcohol (1-dodecanol) from its secretion may synergistically have a negative impact on the growth of selected groups of microorganisms (bacteria and fungi). Some of the analyzed microbes cohabitate with *L. (P.) punctatus* and are potentially entomopathogenic [9,17]. It is yet to be examined whether the synergistic or individual effect of the compounds is responsible for antimicrobial features of the pygidial gland secretion [9]. Considering similar habitat preferences of *L. (A.) cavicola*, it might be possible that certain compounds of its pygidial gland secretion exhibit antimicrobial properties and serve for protection against microbial pathogens.

Pygidial glands are of great significance to ground beetles. They play an important role in protecting these insects from predators [1,11]. This especially applies to the subterranean species of ground beetles, whose survival in inhospitable cave environments is made possible thanks to the action of these glands and other defensive mechanisms [15].

**Author Contributions:** Conceptualization, S.Ć.; methodology, L.V., N.V. and D.R.; software, N.V.; validation, S.Ć., N.V. and L.V.; formal analysis, L.V., M.T. and N.V.; investigation, S.Ć. and M.P.; resources, S.V.; data curation, S.V.; writing—original draft preparation, S.V.; writing—review and editing, S.Ć., L.V. and N.V.; visualization, N.V.; supervision, S.Ć.; project administration, S.Ć.; funding acquisition, S.Ć. All authors have read and agreed to the published version of the manuscript.

**Funding:** This research was funded by the Serbian Ministry of Education, Science and Technological Development (Contracts Nos. 451-03-68/2022-14/200178, 451-03-68/2022-14/200168 and 451-03-68/2022-14/200160), as well as by the Montenegrin Academy of Sciences and Arts (Grant "Catalogue of Pseudoscorpions of Montenegro").

**Institutional Review Board Statement:** Not applicable.

**Data Availability Statement:** Not applicable.

**Acknowledgments:** We would like to thank Boban Rakić (Serbian Institute of Occupational Health "Dr Dragomir Karajović", Belgrade, Serbia), who helped us during field research carried out in a few subterranean sites in Montenegro.

**Conflicts of Interest:** The authors declare no conflict of interest.

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
