# Peer review of "On the Diversity of Semiochemicals of the Pygidial Gland Secretions of Subterranean Ground Beetles (Coleoptera: Carabidae)"

_diversity, doi:10.3390/d15020136_

Round 1

Reviewer 1 Report

I consider this study in general to be of potential interest to fellow chemical ecologists in ground beetles.  I will provide a few minor corrections/suggestions. 

In the materials and methods section:

Why do the authors perform a solvent extraction? Solvent extraction is the most widely used method. However, solvent extraction presents additional problems, including the possible loss of volatile compounds or changes in the ratios of compounds. There are more efficient extraction methods such as Solid-phase microextraction (SPME) or Stir bar sorptive extraction (SBSE).

To the correct identification of compounds requires the determination of retention indices or the use of analytical standards.

I suggest correcting figure 2 because it is not shown in the figure.

Author Response

I consider this study in general to be of potential interest to fellow chemical ecologists in ground beetles. I will provide a few minor corrections/suggestions.

Answer: We thank the reviewer for the positive opinion regarding our manuscript, as well as for the useful comments. 

In the materials and methods section:

Why do the authors perform a solvent extraction? Solvent extraction is the most widely used method. However, solvent extraction presents additional problems, including the possible loss of volatile compounds or changes in the ratios of compounds. There are more efficient extraction methods such as Solid-phase microextraction (SPME) or Stir bar sorptive extraction (SBSE).

Answer: This question is very important to evaluate, and we appreciate the reviewer’s suggestions. We were milking beetles in a vial with solvent rider to soak the secretion directly in the solvent. We are aware that this could affect the volatile profile, but it allowed us to avoid the extraction of epicuticular compounds, and it corresponds with mass literature data if compared with more advanced techniques that the reviewer mentioned. Nevertheless, SPME and SBSE could be very useful if we heat or soak a beetle in the solvent. None of them will be allowed to conduct if we want to keep individuals alive. Moreover, we are milking beetles by squeezing the tip of their abdomen and by pinching the legs with a tweezers, and with this technique we obtain sufficient amounts of extracted compounds. Furthermore, we don’t have access to automated SPME or SBSE (directly connected with GC-MS instrument), but could use only manual SPME and modified SBSE. Both manual SPME and sorptive extraction also have some reproductive issues compared with automated ones. In conclusion, we are willing to compare manual SPME with the milking technique for some of the widely distributed species in future before we switch our sample preparation to new methodology.

To the correct identification of compounds requires the determination of retention indices or the use of analytical standards.

Answer: We completely agree with the reviewer. GC-MS RI data were originally recorded, but not included in the manuscript. To support identification, we insert RI data to L 131-135 and 237-239, as well as to Table 1 (both obtained and literature RI data for standard polar column).

I suggest correcting figure 2 because it is not shown in the figure.

Answer: It is not entirely clear to us what is not shown in the figure. If the reviewer meant the appearance of the peak height of formic acid on the chromatogram (one of the comments of the Reviewer 2), see our answer  within the response to the comments of the Reviewer 2. In conclusion, we insert a new sentence according to this comment (see L 136-139).

Reviewer 2 Report

The authors extracted and analysed the pygidial glands from 8 individual ground beetles and compared their chemical composition and morphological size to other subterranean species. The methodology seems sounded, although I have few remarks regarding the chemical analysis:

Based on figure 2 and the graphical abstract, I am surprised to read that formic acid is the most abundant compound, as it does not appear to be the highest compound on their chromatograms. Since the peak has quite a large base, I wonder if there is another compound co-eluting at the same time. Can the authors re-run the sample on another column to make sure that the identification of this peak is correct or provide the MS spectrum of that peak. Also, since they have identified few compounds, could they confirm all of them with synthetics? If not, can they provide the % of matching with the NIST library for each of them.

Since authors did not used standards, they only calculate relative quantities and did not quantify the chemical compounds as mentioned line 95.

I found the discussion too long and suggest authors to present the results from their comparison analysis and the two tables within the result section and only discuss the ecological habitat influence as they suggested at the end of their introduction. Also, can authors finish their discussion with a concluding sentence, maybe something about the importance of these defensive glands for their survival or conservation?, same for the end of the abstract.

Other minor comments:

-          Could you provide a higher quality image for the abstract?

-          L 57: can you explain why the pygidial gland is regarded as an allomone?

-          L113, can feeding beetle with earthworms influenced the gland content?

-          L117, swap the order of the words: “each individual beetle”..

-          L150: can you back up your claim of no differences between the sexes by stat?     

Author Response

The authors extracted and analysed the pygidial glands from 8 individual ground beetles and compared their chemical composition and morphological size to other subterranean species. The methodology seems sounded, although I have few remarks regarding the chemical analysis:

Answer: We would like to thank the reviewer for the detailed review of the manuscript and useful comments and suggestions.

Based on figure 2 and the graphical abstract, I am surprised to read that formic acid is the most abundant compound, as it does not appear to be the highest compound on their chromatograms.

Answer: We agree with the reviewer that formic acid appears lower in the chromatogram than its concentration suggests. To clarify this, an explanation of the difference in measurement of peak heights and areas was inserted to L 136-139.

Since the peak has quite a large base, I wonder if there is another compound co-eluting at the same time. Can the authors re-run the sample on another column to make sure that the identification of this peak is correct or provide the MS spectrum of that peak.

Answer: We understand the importance of this comment and the possibility of co-eluting in GC-MS chromatograms. Unfortunately, we are not able to re-run samples due to problems related to the age of the samples. Also, we are not able to get new samples because the analysed species doesn’t live in our country. Moreover, formic acid is highly demanding compound for identification using GC methods based on non-polar columns (columns with different polarity), mostly because of the low RI (co-elutes with solvents) and high polarity (needs derivatization and co-elutes with most derivatization reagents). Nevertheless, we analysed the peak of formic acid in details, not just by using its average mass spectrum, but we manually analysed each mass spectrum within this broad peak. There are no additional ions that appear within the peak of formic acid and all extracted ions have exactly the same peak shape. Our findings are confirmed by AMDIS deconvolution analysis as well.

Also, since they have identified few compounds, could they confirm all of them with synthetics? If not, can they provide the % of matching with the NIST library for each of them.

Answer: We agree with the reviewer that identification should be improved.

To support identification, we include RI data and compared them with available literature (NIST Chemistry WebBook and PubChem). Those clarifications are inserted to L 131-135 and 237-239, as well as to Table 1 (both obtained and literature RI data for standard polar column). Furthermore, NIST MS match factors for all compounds are extremely high (at least 940 out of 1,000), but we think that this fact should not be specially mentioned in the manuscript. There is no RI data on the standard polar column for compound 4 (dodecyl formate with a relative intensity of 0.8%) and therefore it is not included in Table 1. Unfortunately, we are not able to run compound standards in the near future to confirm identification.

Nevertheless, here is MS data comparison of peak 4 (Rt 13.81 min and RI 1858) with the NIST 17 library spectrum of dodecyl formate (shown in the attached pdf), with the match factor of 940 out of 1,000 and the reverse match factor of 948 out of 1,000:

In addition to the accurate identification of dodecyl formate, the second and third best matched compounds are 1-dodecanol and cyclododecane. Those two MS spectra could lead to misidentification, but their RIs on standard polar columns are known and they are not related to our identification:

 1-Dodecanol has an RI (for standard polar column) ranging from 1919 to 1984 (https://pubchem.ncbi.nlm.nih.gov/compound/dodecanol#section=Kovats-Retention-Index) and our obtained RI was 1858, which is quite different and could not support the identification.

Cyclododecane has an RI (for standard polar column) ranging from 1497 to1549 (https://pubchem.ncbi.nlm.nih.gov/compound/Cyclododecane#section=Kovats-Retention-Index) and it is completely different.

Since authors did not used standards, they only calculate relative quantities and did not quantify the chemical compounds as mentioned line 95.

Answer: It was corrected as suggested by the reviewer (see L 98).

I found the discussion too long and suggest authors to present the results from their comparison analysis and the two tables within the result section and only discuss the ecological habitat influence as they suggested at the end of their introduction. Also, can authors finish their discussion with a concluding sentence, maybe something about the importance of these defensive glands for their survival or conservation?, same for the end of the abstract.

Answer: We agree with the reviewer that the Discussion is rather long. As the reviewer suggested, we have moved the comparative analysis and Tables 2 and 3 from the Discussion to the Results. The Discussion is significantly shortened and contains only information on the ecological role and significance of the pygidial glands and their secretions. We have added data on the importance of pygidial glands at the end of the Abstract and Discussion.

Other minor comments:

- Could you provide a higher quality image for the abstract?

Answer: We improved the graphical abstract as requested by the reviewer.

- L 57: can you explain why the pygidial gland is regarded as an allomone?

Answer: We have added an explanation of why pygidial gland secretions are considered allomones.

- L113, can feeding beetle with earthworms influenced the gland content?

Answer: We used earthworms which were collected from the same locality to feed ground beetle specimens. Earthworms are one of the most common food sources for ground beetles. So far, there are no known cases of food having an effect on the chemical composition of the pygidial gland secretions of ground beetles.

- L117, swap the order of the words: “each individual beetle”.

Answer: It was corrected as suggested by the reviewer.

- L150: can you back up your claim of no differences between the sexes by stat?

Answer: Too few specimens (4) of each sex were studied, so we were not able to conduct a valid statistical analysis. Analysis of the chemical composition of the pygidial gland secretion of each sex revealed no differences between them, being either qualitative or quantitative. We are not able to obtain new samples in the near future that could enable a valid statistical analysis because the species doesn’t live in our country.
